# Sustainable Synthesis, Antiproliferative and Acetylcholinesterase Inhibition of 1,4- and 1,2-Naphthoquinone Derivatives

**DOI:** 10.3390/molecules28031232

**Published:** 2023-01-27

**Authors:** Rafaela G. Cabral, Gonçalo Viegas, Rita Pacheco, Ana Catarina Sousa, Maria Paula Robalo

**Affiliations:** 1Departamento de Engenharia Química, Instituto Superior de Engenharia de Lisboa, Instituto Politécnico de Lisboa, 1959-007 Lisboa, Portugal; 2Centro de Química Estrutural, Institute of Molecular Sciences, Instituto Superior Técnico, Universidade de Lisboa, 1049-001 Lisboa, Portugal

**Keywords:** naphthoquinones, electrochemistry behavior, acetylcholinesterase inhibitory activity, cytotoxic activity, biological activity

## Abstract

This work describes the design, sustainable synthesis, evaluation of electrochemical and biological properties against HepG2 cell lines, and AChE enzymes of different substituted derivatives of 1,4- and 1,2-naphthoquinones (NQ). A microwave-assisted protocol was optimized with success for the synthesis of the 2-substituted-1,4-NQ series and extended to the 4-substituted-1,2-NQ family, providing an alternative and more sustainable approach to the synthesis of naphthoquinones. The electrochemical properties were studied by cyclic voltammetry, and the redox potentials related to the molecular structural characteristics and the biological properties. Compounds were tested for their potential anti-cancer activity against a hepatocellular carcinoma cell line, HepG2, using MTT assay, and 1,2-NQ derivatives were found to be more active than their 1,4-NQ homologues (**3a**–**f**), with the highest cytotoxic potential found for compound **4a** (EC_50_ = 3 μM). The same trend was found for the inhibitory action against acetylcholinesterase, with 1,2-NQ derivatives showing higher inhibition_50µM_ than their 1,4-NQ homologues, with **4h** being the most potent compound (Inhibition_50µM_ = 85%). Docking studies were performed for the 1,2-NQ derivatives with the highest inhibitions, showing dual binding interactions with both CAS and PAS sites, while the less active 1,4-NQ derivatives showed interactions with PAS and the mid-gorge region.

## 1. Introduction

Naphthoquinones (NQ) have played a crucial role in the drug discovery process been widely used as important pharmacophores or intermediates for the synthesis of bioactive compounds. These heterocyclic compounds are known to exhibit a diverse range of biological properties, such as antimicrobials, antitumoral, antioxidants, antimalarial, and neuroprotectants, among others [1,2,3]. Their mechanisms of action are often related to their redox properties, as most have the capacity to accept one or two electrons to generate highly reactive radical species capable of interacting with biological molecules such as DNA, enzymes, and other proteins [4,5,6]. The electrochemical behavior of these compounds seems to be related to the biological activity and quinone scaffold, with multiple redox functionalities, justifying the continuous interest and efforts to find derivatives with different functionalization and screening for potential biological properties [6,7,8,9].

Different types of cancers and several neurodegenerative disorders are often associated with intracellular redox imbalance and consequent oxidative damage [10,11], making NQ appellative compounds for the redox medicine field. Their anticancer activity has attracted widespread attention, and several natural and synthetic naphthoquinones derivatives have been used in cancer therapy, demonstrating to inhibit the proliferation and induce apoptosis in cancer cells [10,11,12,13,14]. Other works that validate NQ action as selective enzyme inhibitors [15] also showed the potential of using 1,4-NQ and 1,2-NQ derivatives in different therapeutics. Recently, Shin et al. [16] have demonstrated that 1,4-NQ derivatives exhibit neuroprotective effects and may play a key role in the therapeutic strategy against Alzheimer’s disease (AD). Moreover, plant extracts rich in natural naphthoquinones, and synthetic naphthoquinone derivatives were reported as acetylcholinesterase inhibitors (AChEIs) [17,18,19,20,21,22]. The acetylcholinesterase enzyme (AChE, EC 3.1.1.7) plays a major role in the activity of the central and peripheral nervous systems and is one of the main targets of drug therapies for AD [23]. The drugs currently available, such as the AChEIs donepezil, galantamine, and rivastigmine, used in the symptomatic treatment of AD, show non-negligible adverse side effects [24] and the growing importance of neurodegenerative diseases due to the increase in average life expectancy, justifies the search for new alternative leads and the evaluation of the potential of NQ moiety.

The NQ scaffold is undoubtedly important, and the quinone’s reactivity can be modulated by introducing atoms or groups into the central naphthoquinone ring, thus encouraging the design of derivatives and the exploration of their electrochemical and biological properties. Amine groups have been reported to play a key role in many bioactive compounds, and literature data have shown that their presence in the quinone backbone tunes its redox properties and induces oxidative stress in cells and DNA alkylation [25]. Aliphatic amine groups are ubiquitously present in a wide range of compounds with proven biological activities [26,27]. Based on these findings, we design new derivatives of 1,4-NQ and 1,2-NQ, where the combination with different amine substituents aims to promote synergistic effects of both moieties.

The NQ derivatives can be prepared by conventional methods with satisfactory yields, although in most cases, with time-consuming protocols and tedious workups. To overcome these restrictions, new, greener synthetic approaches have been explored [28,29]. Microwave-assisted (MA) organic synthesis is considered an important green approach and, nowadays, is widely used in the pharmaceutical industry for the oriented synthesis of target compounds [30,31], allowing for more energy-efficient and cleaner reactions, the choice of green solvents, time-efficient routes, and higher yields. In this context, and as a part of our interest in developing sustainable synthetic protocols, we report herein the synthesis of several 2-substituted-1,4-NQ and 4-substituted-1,2-NQ derivatives by microwave-assisted protocols, the study of their electrochemical behavior and evaluation of their biological potential, namely, as AChE inhibitors and cytotoxic agents in human hepatocarcinoma HepG2 cell line.

## 2. Results and Discussion

### 2.1. Synthesis and Characterization

A set of compounds derived from 1,4-NQ and 1,2-NQ were studied (Figure 1) to evaluate the viability of an alternative MA protocol for their synthesis, their biological potential, and the impact of the association of these moieties in the same molecule. Aliphatic amines and substituted anilines with electron donor and acceptor groups have been used as nucleophilic partners since it is known that the electronic nature and position of the substituents in the NQ aromatic rings affect their redox potential and, consequently, the associated biological activities [6]. 

The synthesis of 2-substituted-1,4-naphthoquinone derivatives (**3a**–**f**) was performed by microwave-assisted reactions in the presence of a basic catalyst with moderate to excellent yields. The same approach was used for the 4-substituted-1,2-NQ derivatives (**4e,f** and **4h**) in the optimized experimental conditions. The 4-substituted-1,2-NQ derivatives (**4a**–**d** and **4g**) were obtained as previously reported by a biocatalytic strategy using CotA-laccase from *Bacillus subtilis* [32].

The experimental conditions for the MA reactions were optimized by an initial screening using equimolar quantities (50:50 mM) of 1,4-NQ (**1**) and 1,4-phenylenediamine (1,4-PDA) as a nucleophilic agent. A power of 200 W and a maximum temperature of 150 °C were set as microwave conditions, and experimental parameters such as different organic solvents, reaction times, and different catalysts were tested. The reactions were followed by HPLC (Appendix A), and the obtained conversion yields for **3a** are shown in Table 1. 

The reaction between 1,4-NQ (**1**) and 1,4-PDA yields **3a** (60%) after 10 min, in the absence of solvent (entry 2), and the introduction of different solvents (ethanol, acetonitrile, and acetone) did not improve the reaction yields (entries 4–8). Moreover, the use of a basic catalyst, CaCO_3_, as a strategy to increase the nucleophilicity of the arylamine did not improve the yield (entry 3). However, the concomitant use of an organic solvent and a basic catalyst showed to be effective in the improvement of the reaction yield, particularly in the case of acetonitrile and CaCO_3_ (entries 13–15). Other basic catalysts such as NaHCO_3_ and K_2_CO_3_ were tested, in the presence and absence of solvent, without any product formation, as they cause degradation of 1,4-NQ starting material.

From these results, the following optimized experimental conditions were chosen: 5 min under 200 W microwave irradiation, at 150 °C, in 50:50 mM of 1,4-NQ:amine coupler with 1 eq. of CaCO_3_, in acetonitrile:water (1:1), and different arylamines, namely, 4-aminophenol (4-AP), 4-methoxyaniline (4-MA) and 4-aminobenzonitrile (4-ABN) and the alkylamines, butylamine (BA) and diethylamine (DA) were selected, as nucleophiles, to prepare, by Michael 1,4-addition to position C-2, the analogues 2-arylamine-1,4-NQ (**3a**–**d**) and 2-alkylamine-1,4-NQ (**3e**–**f**), as shown in Figure 1.

The performance of the MA approach to the synthesis of derivatives **3a**–**f** was compared with conventional heating under similar experimental conditions (Table 2). Under MA conditions, good conversion yields (57–80%) were obtained for compounds **3a**–**c,** with the electron donor p-substituents enhancing the nucleophilicity of the aromatic amine. The yields are similar to or higher than the conventional reflux in comparatively shorter reaction times (Table 2), demonstrating that the effect of microwave irradiation is not purely thermal. As expected, due to the withdrawing nature of the CN group, compound **3d** was obtained in a low yield (7%). The same approach was used for the alkylamines (BA and DA), but the presence of CaCO_3_ was revealed to have a negative effect, and compounds **3e**–**f** were synthesized in the absence of a catalyst (Table 2). Alkyl amines are more basic than aromatic amines, which justifies the good yields obtained for compounds **3e**–**f** in the absence of a catalyst (63 and 74%, respectively).

The optimized MA conditions were used to prepare the 1,2-NQ derivatives (**4e**, **4f**, and **4h**) with the aliphatic amines and 2-aminophenol (2-AP), respectively (57–75%) (Figure 1), showing that this approach can also be used to prepare this 4-substituted-1,2-NQ series. This last compound was prepared as a proof of concept to understand the influence of the position of the OH group in the aromatic amine.

The structures of synthesized compounds were assigned based on their ^1^H and ^13^C NMR spectra and two-dimensional NMR (HSQC and HMBC) techniques. The ^1^H NMR spectra of compounds **3a**–**f**, **4e**–**f,** and **4h** displayed comparable patterns whose signals multiplicity is in accordance with the proposed structures. Five characteristic signals described the naphthalene scaffold, two duplets (δ = 7.98−8.31 ppm), two triplets (δ = 7.59−7.82 ppm), and one singlet (δ = 5.68−6.22 ppm). The arylamine moieties on compounds **3a**–**d** are characterized by two duplets with resonances in the aromatic region (δ = 6.81−7.00 ppm) and for compound **4h** by two multiplets (δ = 6.90−7.40 ppm). For compounds **3e**–**f** and **4e**–**f**, with aliphatic amine substituents, the ^1^H NMR spectra showed the characteristic signals in the aliphatic region (δ = 0.98–3.62 ppm). In the ^13^C NMR spectra, all carbon resonances were detected in the expected regions and assigned based on their chemical shifts and HSQC and HMBC cross signals. Furthermore, the structures were confirmed by ESI-MS spectrometry in accordance with their molecular formula. The mass spectra of all products exhibit the correspondent signals to the protonated molecules [M + H]^+^ at *m/z* 265, 266, 280, 275, 266 for **3a**–**d** and **4h** and *m/z* 230 for **3e,f** and **4e,f,** respectively. Generally, for all ions, the first fragmentation pattern [ESI(+)-MS^2^] is consistent with the correspondent typical loss of CO leading to signals [M+1−CO]^+^.

### 2.2. Electrochemistry Studies

Electron transfers play important roles in the bioactivation of redox-active drugs, in their metabolism/catabolism, and in their targeted release at precise destinations and frequently promoted their ligand-target interactions. Redox potentials provide information on the feasibility of electron transfer in vivo and established relationships between the ease of reduction (represented by E_pc_ or E_1/2_) and biological activities, demonstrating the relevance of electrochemical studies as tools for the comprehension of drug mechanism of action against various diseases, for the prediction of biological activities and for the design of potentially active compounds [6,33,34,35]. 

To obtain an insight into the electronic character of the studied naphthoquinone derivatives and to find out possible relationships with their potential biological activities, the electrochemical behavior of 1,4-NQ (**1**) and 1,2-NQ (**2**) and their 2-substituted-1,4-NQ (**3a**–**f**) and 4-substituted-1,2-NQ derivatives (**4a**–**h**), respectively, were studied by cyclic voltammetry (CV) in aprotic medium (CH_3_CN + TBAP, 0.1 mol.L^−^^1^) and the electrochemical selected data are listed in Table 3 (for complete electrochemical data, see Appendix A).

The cyclic voltammogram of the precursor, 1,4-NQ (**1**) (Figure 2a), shows the presence of two sequential mono-electronic reductive waves, being the first one quasi-reversible (E_pc_(I) = −0.70 V vs. SCE, E_1/2_ = −0.66 V) and the second, broader and less-defined, irreversible (E_pc_(II)= −1.36 V). In order to estimate the contribution of the relative position of the quinonoid groups to the electrochemical response, the 1,2-NQ (**2**) was also studied in the same conditions (Figure 2b). Similarly, the 1,2-NQ (**2**) shows two sequential mono-electronic quasi-reversible waves (E_pc_(I) = −0.55 V, E_1/2_ = −0.50 V; E_pc_(II) = −0.98 V). They correspond, as already known, the first pair to the generation of the semiquinone (SNQ^•−^), followed, in the second step, by the dianion (NQ^2−^) formation [6,7,8]. Thus, 1,2-NQ is easier to be reduced than 1,4-NQ being this behavior is observed in both reductive steps. 

The influence of substituents on the electrochemical properties of 1,4-NQ and 1,2-NQ was also studied for both series and the CV of **3a**, the most cytotoxic 1,4-NQ derivative is shown, as an example, in Figure 2c. The electrochemical data of the other 2-substituted−1,4-NQ derivatives are similar (data in Table 3). The redox profile showed the following two independent systems: the reduction of the naphthoquinone moiety and the oxidation of the amine substituent moiety in the anodic part (Figure 2c). The redox processes displayed in the range +0.72 to +1.55 V were related to oxidations within the substituted anilines or aliphatic amines. The substitution at the 1,4-NQ moiety generally leads to easier oxidations when compared with the correspondent 1,2-NQ. Analysis of the cathodic region showed one or two reductive processes for the 1,4-NQ and 1,2-NQ nucleus, shifted to lower potential values and less reversible upon substitution with the aromatic or aliphatic amines. The ease of reduction, represented by the potential of the first well-defined wave (E_pc_(I)), for both NQ series follows the following order: CN > OH ~ NHPh > OCH_3_ ~ NH_2_ > Dietylamine ~ Butilamine. In the 1,4-NQ series, a difference of 240 mV was observed between compound **3d**, the most easily reduced, and the butylamine derivative (**3e**), reduced at more negative potentials. For the 1,2-NQ series, the difference is even higher (310 mV) between **4d** and **4e**. Generally, the trend observed on the redox potentials is clearly related to the electronic effects of the substituents. Aliphatic amine substituents **3e**, **3f**, **4e**, and **4f** exhibit the lowest redox potentials indicating that the aliphatic amino groups decreased the electron affinity of the naphthoquinone scaffolds. Electron donors or withdrawing substituents within the amine aromatic ring directly influence the electron density of the 1,2- and 1,4-NQs moieties. While 1,4-NQ or 1,2-NQ with electron-donating substituents such as NH_2_, OH, and OCH_3_ have comparatively low values, the electron-withdrawing CN shows no appreciable change, being, therefore, easier to reduce.

### 2.3. In Silico Studies

As a first approach, the Molinspiration cheminformatics program [36] was used to predict the in silico molecular properties of the 2-substituted-1,4-NQ and 4-substituted-1,2-NQ derivatives. The theoretical oral bioavailability was predicted using the “Rule of five” proposed by Lipinski and Lombardo [37] and the percentage of oral absorption (ABS%) [38] (see Appendix A). The bioactivity scores were estimated for drug targets, such as G-protein coupled receptor ligands, ion channel modulators, kinase inhibitors, nuclear receptor ligands, protease inhibitors, and enzyme inhibitors, and results were summarized in Appendix A.

Based on the results, all compounds showed adequate molecular properties and met the criteria established by Lipinski’s “rule of five”, presenting adequate hydrophobic properties (logP ≤ 5, exception for compound **4g**), size properties (M ≤ 500 g/mol and TPSA ≤ 140 Å) and bond restrictions (n_OH_ ≤ 10 and n_OHNH_ ≤ 5). Furthermore, the calculated ABS% (from 84.6 to 99.4% range) suggests good membrane permeability for the compounds. Regarding the predicted bioactivity scores against common molecular targets, the results clearly indicate an improvement in the bioactivity scores upon the substitution of 1,2-NQ and 1,4-NQ scaffolds. In particular, the results showed that both 2-aryl-1,4-NQs and 4-aryl-1,2-NQs are expected to be biologically active as kinase inhibitors (see Appendix A). In fact, the 1,4-NQ skeleton and other quinone-derived compounds had already been reported as active against kinases, by biochemical binding and molecular interaction with the kinases catalytic sites [39]. Kinase inhibitors are a large group of antineoplastic agents targeting protein kinases, which are altered in cancer cells leading to abnormal growth; hence, kinase inhibitors are considered a promising approach for anticancer drug development [40].

### 2.4. Biological Assays

#### 2.4.1. In Vitro Cytotoxicity against Human Hepatocarcinoma Cell Line HepG2

The potential in cancer therapy showed by natural and synthetic naphthoquinones derivatives [9,10,11] and the predicted biological activity as kinase inhibitors provided by in silico studies prompted us to determine the cytotoxicity profile of substituted 1,4-NQ and 1,2-NQ derivatives (**3a**–**f** and **4a**–**h**) together with 1,4-NQ and 1,2-NQ. 

A hepatocellular carcinoma cell line HepG2, recognized for studying the metabolism of anticancer drugs, was used [41]. The viability of HepG2 cell line culture after 24 h incubation with the compounds using the MTT (3-[4,5-dimethylthiazol-2-yl]-2,5-diphenyl tetrazolium bromide) assay [42,43] was evaluated, and doxorubicin (**Doxo**) was used as a positive control. A dose-dependent cytotoxic effect was observed for all compounds (see Appendix A), the effective concentration of each compound (μM) leading to the death of 50% of the cells (EC_50_) was calculated, and the results are presented in Figure 3 (see also Appendix A).

All compounds showed cytotoxic activity in the μM range and inhibited cell proliferation of the HepG2 cell line. These results additionally showed that 1,2-NQ derivatives (**4a**–**f**) exhibit a stronger effect than their 1,4-NQ homologues (**3a**–**f**). Derivatization with 4-substituted aromatic amines with donor groups leads to an improvement in the cytotoxic potential, with EC_50_ values ranging from 9 to 21 µM for **3a**–**c** and from 3 to 7 µM for **4a**–**c** when compared to their respective precursors 1,4-NQ (EC_50_ = 32 µM) and 1,2-NQ (EC_50_ = 26 µM), respectively. Derivatization with 4-substituted aromatic amine with a cyano group (**3d** and **4d**) and with N-alkyl amines (**3e**–**f** and **4e**–**f**) decreases the cytotoxicity for both precursors. The highest cytotoxic activity was found for compound **4a** (EC_50_ = 3 μM), which is a promising result when compared to doxorubicin (EC_50_ = 1.4 μM) used as a positive control.

The cytotoxic activity of these compounds can be related to their electrochemical properties, for which the experimental data showed that overall, the 1,2-NQ derivatives present higher reductive potentials (E_pc_) than their 1,4-NQ homologues and consequently a stronger oxidizing power, which may justify its cytotoxic effect. It has been postulated that the cytotoxicity is mediated through semiquinone radical species, which are formed during the quinone conversion and vice-versa and thus is highly dependent on the electrochemical potential of the initially formed radical species [34,44]. The anticancer agent doxorubicin and other quinones are also known to cause chemical and oxidative damage to cells [45]. 

#### 2.4.2. Inhibition of Acetylcholinesterase Activity

The acetylcholinesterase enzyme (AChE, EC 3.1.1.7) plays a major role in the activity of the central and peripheral nervous systems, and its inhibition represents an important therapeutic target for AD [23]. The adverse effects shown by AChEIs in use in the treatment of Alzheimer’s and dementia symptoms [24], the growing importance of neurodegenerative diseases, and previous studies on naphthoquinones as AchEIs [21] led us to extend our studies to the potential for inhibition of AchE activity. 

The inhibitory activities against AchE were evaluated for all compounds at a concentration of 50 µM, and the Inhibition_50µM_ values are given in Figure 4 (see also Appendix A).

Results show that 1,4-NQ (**1**) and 1,2-NQ (**2**) exhibit AChE inhibitory values for a concentration of 50 µM of 51% and 35%, respectively. As shown in Figure 4, different trends are observed after substitution with aryl- or alkylamines. The 2-substituted-1,4-NQ derivatives **(3a**–**f**) decrease the 1,4-NQ inhibitory activity against AChE (Figure 4a), whereas 4-substituted-1,2-NQ derivatives (**4a**–**f**) showed an increase in the potential shown by 1,2-NQ (Figure 4b). Thus, the 1,2-NQ derivatives (**4a**–**f**) showed moderate inhibition to AChE, with compound **4h** presenting the highest AChE inhibitory activity (Inhibition_50µM_ = 85%). In fact, when the position of the hydroxyl group changes from para (**4b**) to ortho (**4h**) position, a slight increase in the inhibitory activity against AChE was observed. 

Additionally, a dose-dependent AChE inhibition effect was determined for all, and the effective concentration of each compound (μM) that leads to the 50% of enzyme inhibition (AChEIC_50_) was calculated for compounds **4b** (30 μM) and **4h** (16 μM). These values, higher than the AChEIC_50_ values of the reference AChE inhibitors rivastigmine (1.03 μM) or galantamine (1.99 μM) [46], show that these compounds are only moderate AChE inhibitors and further structural changes must be considered for these 1,2-NQ and 1,4-NQ families can be foreseen as alternative drug scaffolds.

### 2.5. Molecular Docking Studies

Molecular docking studies were performed to further elucidate the binding interactions between the 1,2-NQ derivative **4b**, the most potent AChE inhibitor, and the corresponding 1,4-NQ derivative **3b**, with poorer performance, and AChE enzyme active site gorge. The molecular docking studies were performed with AutoDock Vina v.1.2.0 software. Based on the docking score, the best conformation of the compounds binding at the AChE active site gorge was chosen, and the interactions of 1,4-NQ derivative **3b** and 1,2-NQ derivative **4b** with AChE are shown in Figure 5. Bond types and distances are summarized in Appendix A.

The docking results showed that both compounds fit well in the binding pocket of AChE, and hydrogen bond and π-π stacking interactions were found between both compounds and the amino acid residues at the AChE gorge. As depicted in Figure 5, the 1,4-NQ derivative **3b** shows one hydrogen bond between one quinonoid carbonyl group and the Phe295 residue located at the mid-gorge. Both aromatic rings of the 1,4-NQ scaffold stack with the aromatic residues, Trp286 and Tyr341, belong to the peripheral anionic site (PAS), which is situated near the entrance of the gorge. Moreover, the NQ ring of **3b** also showed an additional hydrogen bond with Val294 located at the entrance of the AChE active site. Both the hydroxyl group and the peripheric aromatic ring of **3b** were not involved in any interactions. A molecular docking study of the 1,2-NQ derivative **4b** (Figure 5) showed a hydrogen bond involving the quinonoid carbonyl group and Tyr124 residue located at the PAS region. In addition, the Tyr341 residue at PAS region interacts via arene-arene stacking with **4b**. Other strong hydrogen bonds were observed between the hydroxyl group of the peripheric aromatic ring and Tyr133 residue on the catalytic anionic site (CAS) and Glu202, one of the acidic amino acids surrounding the CAS region, located at the bottom of the gorge and near the AChE catalytic triad. Other crucial amino acids at the CAS region, Trp86, and Tyr337 residues, interact via important π-π stacking interactions with all the aromatic rings present on **4b**. The CAS site is involved in the binding of the choline moiety of ACh and the residues Trp86 and Tyr337 were seen to be critical for the inhibition of AChE [47]. 

In general, the major difference between the two compounds, **3b** and **4b**, is related to the fact that the 1,4-NQ derivative (**3b**) showed interactions between the NQ ring, the PAS, and mid-gorge regions, whereas the 1,2-NQ derivative (**4b**) interacts with the PAS region and additionally, by the NQ ring and the peripheric aromatic substituent, with several amino acid residues nearby the CAS region. The importance of these interactions with CAS residues was reinforced by the docking results obtained for derivative **4h**. This 1,2-NQ derivative showed several interactions within the CAS region surrounding residues (Trp86, Tyr133 and Tyr337) both by hydrogen bond and π-π stacking interactions (Appendix A). 

Comparison of the docking results from compounds **4b** and **4h** (Figure 6) evidence that both compounds at the AChE active site interact with residues at PAS and CAS.

This strong hydrogen binding and hydrophobic interaction with the active site residues of AChE may be one of the reasons for the good inhibitory activity shown by the 1,2-NQ derivative compounds in the series. Dual-binding cholinesterase inhibitors, such as donepezil, have been reported to interact with both CAS and PAS regions, and these interactions were reported to be related to their potent AChE inhibition profiles [47,48,49].

## 3. Materials and Methods

Microwave-assisted syntheses were performed with a CEM Discover SP microwave synthesizer system (2.45 GHz, 300 W, CEM Microwave Technology Ltd., Buckingham, UK) equipped with a non-contact infrared temperature sensor. Experiments were performed in sealed Pyrex microwave vials using dynamic mode and the reaction temperature was controlled by variable microwave irradiation (0–200 W) and cooling by nitrogen current. 

Reagents and solvents were purchased from Merck (Darmstadt, Germany) or Alfa Aesar (Kandel, Germany) with high purity and used without further purification. Analytical thin-layer chromatography (TLC) was performed using silica gel 60 F254 precoated plates (0.25 mm thickness) with a fluorescent indicator. Column chromatography was carried out using silica gel 60 (EM Separations Technology) as the stationary phase. ^1^H and ^13^C NMR spectra were obtained in CD_3_OD-d_4_ as solvent using a Bruker Avance 400 MHz spectrometer (Bruker, Billerica, MA, USA) with a 5 mm probe. Chemical shifts (δ) are given in ppm and reported based on TMS as standard. Spin multiplicities are given as s (singlet), d (doublet), t (triplet), and m (multiplet)). Coupling constants (J) are in Hz. Spectra are reported as follows: chemical shift (δ, ppm), multiplicity, integration, and coupling constants (Hz).

HPLC was conducted on a Waters Alliance e2695 Separations Module (Massachusetts, EUA) with 2489 UV/Vis Detector, also from Waters. The chromatographic separation was conducted with a Symmetry C18 column (250 × 4.6 mm, 5 μm; Waters), with formic acid 0.1% (*v*/*v*) and acetonitrile as mobile phases A and B, respectively. A 32-min linear gradient from 5 to 70% of acetonitrile mobile phase B, followed by an 8-min linear gradient to 100% of mobile phase B, was used in all instances. The injection volume was 10 µL, and the detection was performed at 270 nm.

Low-resolution mass spectra were recorded on an LCQ Fleet, Thermo Scientific Ion Trap mass spectrometer, operated in the electrospray ionization (ESI) positive/negative ion modes. The optimized parameters were as follows: ion spray voltage, ±4.5 kV; capillary voltage, +16 and −20 V; tube lens offset, −63 and +82 V; sheath gas (N2), 80 arbitrary units; auxiliary gas, 5 arbitrary units; capillary temperature, 250 °C. The spectra were recorded in the range of 50–1000 Da. Spectrum typically corresponds to the average of 20–35 scans. Sequential mass spectra were obtained with an isolation window of 2 Da, a 25–30% relative collision energy and with an activation energy of 30 msec. High-resolution ESI (+/−) mass spectra were obtained on a QTOF Impact IITM mass spectrometer (Bruker Daltonics, GMBH; Germany), operating in the high-resolution ion mode. Calibration of the TOF analyzer was performed with a 10 mM sodium formate calibrant solution. Data were processed using Data Analysis 4.2 software.

The redox potentials were measured by cyclic voltammetry using an EG&G Princeton Applied Research (PAR) Model 273A potentiostat/galvanostat monitored by a personal computer with the Electrochemistry PowerSuite v2.51 software from PAR. Cyclic voltammograms were obtained in 1 mM of compounds in anhydrous acetonitrile solutions containing tetrabutylammonium hexafluorophosphate (TBAPF_6_) 0.1 M as supporting electrolyte. A three-electrode configuration cell was used with a homemade platinum disk working electrode (1.0 mm diameter) probed by a Luggin capillary connected to a silver-wire pseudo-reference electrode and a platinum wire auxiliary electrode. The potentials were scanned from −1.8 to 1.8 V at a scan rate of 100 mV·s^−^^1^. All measurements were done at room temperature and the solutions were deaerated with nitrogen atmosphere before use. All the potentials reported were measured against the ferrocene/ferrocenium redox couple as internal standard and corrected to SCE (using the ferrocenium/ferrocene redox couple E_1/2_ = 0.40 V vs. SCE).

### 3.1. Optimized Conditions for Microwave-Assisted Synthesis of 1,4-NQ Derivatives (***3a***–***d***)

The 1,4-naphthoquinone (**1**) (15.8 mg, 50 mmol) and aryl amines (1,4-PDA, 4-AP, 4-MA, and 4-ABN) (50 mmol) were dissolved in acetonitrile (1mL) in a microwave vial. In total, 1 mL of CaCO_3_ aqueous solution (50 mmol) was added and the vial sealed. The resulting solution was stirred under microwave irradiation (200 W) at 150 °C for 5 min. After cooling, products (**3a–d**) were extracted with ethyl acetate. Solvent was evaporated under reduced pressure and residues purified by column chromatography (silica gel, ethyl acetate:chloroform 1:3). 

*2-(1,4-Phenylenediamine)-1,4-naphthoquinone* (**3a**): Purple solid (80%): ^1^H NMR (400 MHz, MeOD-d_4_): δ(ppm) = 8.11 (d, 1H, *J* = 7.2 Hz, H5), 8.02 (d, 1H, *J* = 7.5 Hz, H8), 7.80 (t, 1H, *J* = 7.8 Hz, H7), 7.72 (t, 1H, *J* = 7.4 Hz, H6), 7.09 (d, 2H, *J* = 8.6 Hz, H12, H16), 6.81 (d, 2H, *J* = 8.6 Hz, H13, H15) and 6.01 (s, 1H, H3). ^13^C NMR (100 MHz, MeOD-d_4_): δ(ppm) = 185.5 (C4), 183.0 (C1), 147.8 (C2), 135.9 (C7), 134.8 (C10), 133.5 (C6), 132.1 (C9), 127.5 (C5), 126.8 (C8), 126.5 (C12, C16), 116.9 (C13, C15) and 101.4 (C3). ESI-MS positive mode: 287 [M+Na]^+^; 265 [M+H]^+^, MS^2^ (*m*/*z*): 237 [M+H-CO]^+^, MS^3^ (*m*/*z*): 209 [M+H-CO-CO]^+^; ESI-MS negative mode: 263 [M−H]^−^. ESI-HRMS: *m*/*z* calcd. for [C_16_H_13_N_2_O_2_+H]^+^: 265.0972; found 265.0980.*2-(4-aminophenol)-1,4-naphthoquinone* (**3b**): orange solid (59%): ^1^H NMR (400 MHz, MeOD-d_4_): δ(ppm) = 8.11 (d, 1H, *J* = 7.7 Hz, H5), 8.02 (d, 1H, *J* = 7.6 Hz, H8), 7.80 (t, 1H, *J* = 7.5 Hz, H7), 7.70 (t, 1H, *J* = 7.5 Hz, H6), 7.17 (d, 2H, *J* = 8.8 Hz, H12, H16), 6.87 (d, 2H, *J* = 8.8 Hz, H13, H15) and 6.0 (s, 1H, H3). ^13^C NMR (100 MHz, MeOD-d_4_): δ(ppm) = 185.6 (C4), 182.9 (C1), 157.2 (C14), 135.9 (C7), 133.6 (C6), 132.1 (C10), 130.4 (C9), 127.4 (C5), 126.9 (C12, C16), 126.8 (C8), 117.1 (C13, C15) and 101.7 (C3). ESI-MS positive mode: 266 [M+H]^+^, MS^2^ (*m*/*z*): 238 [M+H-CO]^+^. ESI-HRMS: *m*/*z* calcd. for [C_16_H_11_NO_3_+H]^+^: 266.0812; found 266.0810.*2-(4-methoxyaniline)-1,4-naphthoquinone* (**3c**): brown solid (57%): ^1^H NMR (400 MHz, MeOD-d_4_): δ(ppm) = 8.09 (d, 1H, *J* = 7.6 Hz, H5), 8.00 (d, 1H, *J* = 7.7 Hz, H8), 7.78 (t, 1H, *J* = 7.5 Hz, H7), 7.70 (t, 1H, *J* = 7.5 Hz, H6), 7.25 (d, 2H, *J* = 8.8 Hz, H12, H16), 7.00 (d, 2H, *J* = 8.8 Hz, H13, H15), 6.02 (s, 1H, H3) and 3.82 (s, 3H, CH_3_). ^13^C NMR (100 MHz, MeOD-d_4_): δ(ppm) = 185.6 (C4), 182.8 (C1), 159.4 (C14), 135.8 (C7), 133.6 (C6), 133.2 (C10), 132.1 (C9), 129.4 (C11), 127.4 (C5), 126.8 (C8), 126.7 (C12, C16), 115.8 (C13, C15), 101.9 (C3) and 56.0 (CH_3_). ESI-MS positive mode: 280 [M+H]^+^, MS^2^ (*m/z*): 252 [M+H-CO]^+^. ESI-HRMS: *m/z* calcd. for [C_17_H_13_NO_3_+H]^+^: 280.0968; found 280.0976.*2-(4-aminobenzonitrile)-1,4-naphthoquinone* (**3d**): orange solid (7%): ^1^H NMR (400 MHz, MeOD-d_4_): δ(ppm) = 8.31 (d, 1H, *J* = 7.8 Hz, H5), 8.05 (d, 1H, *J* = 7.8 Hz, H8), 7.82 (t, 1H, *J* = 7.2 Hz, H7), 7.65 (t, 1H, *J* = 7.2 Hz, H6), 7.48 (d, 2H, *J* = 9.0 Hz, H12, H16), 7.35 (d, 2H, *J* = 9.0 Hz, H13, H15), 6.22 (s, 1H, H3). ESI-MS positive mode: 275 [M+H]^+^, MS^2^ (*m*/*z*): 247 [M+H-CO]^+^. ESI-HRMS: *m*/*z* calcd. for [C_17_H_10_N_2_O_2_+H]^+^: 275.0815; found 275.0819.

### 3.2. Optimized Conditions for Microwave Synthesis of 1,4-NQ Derivatives (***3e,f***) and 1,2-NQ Derivatives (***4e,f***)

The 1,4-naphthoquinone (**1**) or 1,2-naphthoquinone (**2**) (15.8 mg, 50 mmol) and alkyl amines (BA and DA) (50 mmol) were dissolved in acetonitrile (2 mL) and the microwave vial was sealed. Solution was stirred under microwave irradiation (200 W) at 150 °C for 5 min. After cooling, products were extracted with ethyl acetate. Solvent was evaporated under reduced pressure and residues purified by column chromatography (silica gel, ethyl acetate:chloroform 1:3). 

*2-(butan-1-amine)-1,4-naphthoquinone* (**3e**): orange solid (63%): ^1^H NMR (400 MHz, MeOD-d_4_): δ(ppm) = 8.01 (dd, 2H, *J* = 6.9 Hz, H5, H8), 7.75 (t, 1H, *J* = 7.5 Hz, H7), 7.66 (t, 1H, *J* = 7.5 Hz, H6), 5.68 (s, 1H, H3), 3.23 (t, 2H, *J* = 7.2 Hz, H11), 1.66 (d, 2H, *J* = 7.4 Hz, H12), 1.44 (d, 2H, *J* = 7.4 Hz, H13) and 0.98 (t, 3H, *J* = 7.3 Hz, H14). ^13^C NMR (100 MHz, MeOD-d_4_): δ(ppm) = 184.7 (C4), 182.6 (C1), 150.8 (C2), 135.8 (C7), 135.0 (C9), 133.3 (C6), 132.1 (C10), 127.3 (C8), 126.8 (C5), 99.9 (C3), 43.2 (C11), 31.1 (C12), 21.3 (C13) 14.1 (C14). ESI-MS positive mode: 230 [M+H]^+^. ESI-HRMS: *m*/*z* calcd. for [C_14_H_15_NO_2_+H]^+^: 230.1176; found 230.1180.*2-(diethylamine)-1,4-naphthoquinone* (**3f**): red solid (74%): ^1^H NMR (400 MHz, MeOD-d_4_): δ(ppm) = 7.98 (dd, 2H, *J* = 7.9 Hz, H5, H8), 7.76 (t, 1H, *J* = 7.4 Hz, H7), 7.68 (t, 1H, *J* = 7.4 Hz, H6), 5.86 (s, 1H, H3), 3.62 (m, 2H, H11) and 1.32 (t, 2H, *J* = 7.0 Hz, H12). ^13^C NMR (100 MHz, MeOD-d_4_): δ(ppm) = 184.7 (C4), 184.4 (C1), 153.3 (C2), 135.0 (C7), 134.3 (C9), 134.1 (C10), 133.3 (C6), 125.9 (C5, C8), 105.0 (C3), 48.0 (C11, C11′) and 12.9 (C12, C12′). ESI-MS positive mode: 230 [M+H]^+^. ESI-HRMS: *m/z* calcd. for [C_14_H_15_NO_2_+H]^+^: 230.1176; found 230.1177.*4-(butan-1-amine)-1,2-naphthoquinone* (**4e**): red solid (72%): ^1^H NMR (400 MHz, MeOD-d_4_): δ(ppm) = 8.12 (d, 1H, *J* = 8.0 Hz, H8), 8.05 (d, 1H, *J* = 8.0 Hz, H5), 7.80 (t, 1H, *J* = 8.0 Hz, H6), 7.70 (t, 1H, *J* = 8.0 Hz, H7), 5.90 (s, 1H, H3), 3.52 (t, 2H, *J* = 8.0 Hz, H11), 1.76 (d, 2H, *J* = 7.4 Hz, H12), 1.48 (d, 2H, *J* = 8.0 Hz, H13) and 1.00 (t, 3H, *J* = 8.0 Hz, H14). ^13^C NMR (100 MHz, MeOD-d_4_): δ(ppm) = 180.9 (C1), 175.3 (C2), 159.4 (C4), 135.8 (C7), 135.6 (C6), 132.9 (C7), 132.6 (C10), 131.9 (C9), 129.3 (C8), 124.6 (C5), 98.9 (C3), 44.9 (C11), 31.5 (C12), 21.3 (C13) 14.1 (C14). ESI-MS positive mode: 256 [M+Na]^+^ and 230 [M+H]^+^. ESI-HRMS: *m/z* calcd. for [C_14_H_15_NO_2_+H]^+^: 230.1176; found 230.1183.*4-(diethylamine)-1,2-naphthoquinone* (**4f**): red solid (75%): ^1^H NMR (400 MHz, MeOD-d_4_): δ(ppm) = 8.05 (dd, 2H, *J* = 7.9 Hz, H5, H8), 7.75 (t, 2H, *J* = 7.4 Hz, H6, H7), 6.02 (s, 1H, H3), 3.68 (q, 4H, H11, H11′) and 1.41 (t, 6H, *J* = 8.0 Hz, H12, H12′). ^13^C NMR (400 MHz, MeOD-d_4_): δ(ppm) = 183.7 (C1), 177.6 (C2), 165.6 (C4), 135.0 (C6), 133.3 (C9), 132.4 (C5), 130.2 (C10), 129.9 (C9), 129.9 (C8), 128.7 (C7), 105.4 (C3), 47.9 (C11, C11′) and 13.0 (C12, C12′). ESI-MS positive mode: ESI-MS positive mode: 256 [M+Na]^+^ and 230 [M+H]^+^. ESI-HRMS: *m/z* calcd. for [C_14_H_15_NO_2_+H]^+^: 230.1176; found 230.1176.

### 3.3. Bioactivity Score Prediction

Drug score values indicate the overall potential of a compound to be a drug candidate. Mol inspiration is a web-based tool used to predict the bioactivity score of synthesized compounds against regular human receptors such as GPCRs, ion channels, kinases, nuclear receptors, proteases, and enzymes [35]. The evaluation of drug likeliness was based on Lipinski’s rule of five.

### 3.4. Biological Activities

#### 3.4.1. Cytotoxicity Assay against HepG2 Human Cell Line 

HepG2 human hepatocyte carcinoma cell line was purchased from European Collection of Authenticated Cell Cultures (ECACC). HepG2 cells (ECACCC# 85011430) were cultured in Dulbecco’s Modified Eagle´s Medium (DMEM) supplemented with 10% fetal bovine serum, antibiotic-antimycotic (100 U/mL penicillin-streptomycin and 0.25 μg amphotericin B) and 2 mM L-glutamine at 37 °C in an atmosphere containing 5% CO_2_. The 3-(4,5-dimethylthiazol-2-yl)-2,5-diphenyltetrazolium bromide (MTT) method, described in Gaspar et al., [50] was used for assaying cell viability. Briefly, the cells were grown in the supplemented medium in 96-well microplates in an incubator with 5% CO_2_ at 37 °C, until reaching 100% confluence. After growth, the medium was replaced by the compounds’ solutions at different concentrations. The compounds were solubilized in supplemented medium containing 0.5% DMSO final concentration. Cells treated only with medium containing 0.5% DMSO served as a negative control. After 24 h incubation, the compound’s solutions were replaced by 100 µL of 0.5 mg/mL MTT solution in DMEM culture medium and incubated at 37 °C, 5% CO_2_ for 2–4 h. The formed formazan crystals were dissolved in 200 µL of methanol and the absorbance at 595 nm was registered against 630 nm (reference wavelength). For each concentration of the compounds, the percentage of growth inhibition/cytotoxicity was evaluated considering 100% of viability for the absorbance of the negative control. Dose-response curves for each compound were plotted using the Graph Pa Prim 4.0 software and the effective concentration leading to the death of 50% of the cells (EC_50_) was calculated. All experiments were replicated at least six times.

#### 3.4.2. AChE Inhibition Assay

The inhibition of acetylcholinesterase (AChE) enzymatic activity was measured using the Ellman’s colorimetric method with some alterations [51]. Briefly, 325 μL of 50 mM Tris–HCl buffer (pH 8), 100 μL of a compound’s solution with 1% DMSO, and 25 μL of AChE (0.1 U/mL) in 50 mM Tris–HCl buffer pH 8 were incubated for 15 min. Subsequently, 75 μL of acetylthiocholine iodide (AChI) (0.023 mg/mL) and 475 μL of 3 mM 5,5′-dithiobis(2-nitrobenzoic acid) (DTNB) in Tris–HCl buffer (pH 8) containing 0.05 M NaCl and 0.021 M MgCl_2_ were added to initiate the reaction. The initial rate of the enzymatic reaction was quantified by measuring the absorbance at 405 nm for 5 min (V[compound]). A control reaction was carried out using 1% DMSO water solution, instead of the compound’s solution, and this initial rate was considered 100% of the enzymatic activity, Vcontrol. The percentage of AChE inhibition (I) for various concentrations of the different compounds was determined as the ratio of V[compound] and Vcontrol. For each compound, both the AChEIC_50_ value (the concentration that inhibits 50% of the AChE activity) and the percentage of inhibition at the compound concentration of 50 µM (Inhibition_50µM_) were calculated by plotting I for each compound concentration. All the assays were carried out in triplicate.

#### 3.4.3. Molecular Docking

Molecular docking was performed using AutoDock Vina v.1.2.0 software, CCSB. Both the compounds and the enzyme (PBD ID: 4EY7) were prepared for docking simulations via DockPrep. This function of Chimera 1.16 software removed water molecules and added hydrogens to the compounds, which structure was obtained through ChemDraw software. Five models were generated for each compound, and the highest-scoring model was viewed and analyzed using the Discovery Studio software (BIOVIA, San Diego, CA, USA).

## 4. Conclusions

A microwave-assisted protocol was successfully established for the synthesis of 2-substituted-1,4-NQ and 4-substituted-1,2-NQ derivatives, with good yields and completing the reactions in shorter times. The electrochemical properties showed to be driven by the reductive processes within the NQ moiety, and the 1,2-NQ series showed to be easier reduced than the corresponding 1,4-NQ series. Moreover, the trend in the redox potential within each series is clearly related to the electronic properties of the amine substituents. 

The potential biological activity of the compounds was assessed for their cytotoxicity against HepG2 cell lines and as AChE inhibitors, showing that 1,2-NQ series exhibit a dual action, both as promising cytotoxic agents and moderate AChE inhibitors. The results showed a dependence on chemical modifications in the NQ moiety with an increase in cytotoxic and AChE inhibitory effects, in particular, for electron-donating substituted aromatic amines. Comparison between biological activities and electrochemical parameters showed that the first wave reduction potential is an important parameter as follows: the most easily reduced naphthoquinones (>E_pc_) were the most active against the HepG2 cancer cell lines and as AChE inhibitors. The observed trend suggests that the bioreduction may play an important role in the biological activity of these compounds, but other factors must also be considered. 

To elucidate the possible mechanism of AChE targeting action, docking simulations were performed for the more active compounds (**4b** and **4h**), confirming that both compounds interact with amino acid residues in the PAS and CAS of the AChE active site through hydrogen binding and hydrophobic interactions. These interactions, also found for dual-binding cholinesterase inhibitors, such as donepezil, may be related to the good inhibition profiles found for the 4-substituted-1,2-NQ family.

## Data Availability

Not applicable.

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
