# Peer review of "Sustainable Synthesis, Antiproliferative and Acetylcholinesterase Inhibition of 1,4- and 1,2-Naphthoquinone Derivatives"

_molecules, 2023, doi:10.3390/molecules28031232_

Round 1

Reviewer 1 Report

In this article, the authors showed a complete analysis of 1,4- and 1,2-naphthoquinone derivatives, presenting synthesis, electrochemical analysis, biological assays, in silico and molecular docking studies. The results indicate that 1,2-naphthoquinone derivatives are potential cytotoxic agents and AChE inhibitors.

·       Scheme 1, in a 3h must be 4h.

·       Line 134, where must be were

·       Table 2, check conventional heating values for 3d

·    Line 155, I suggest changing characterized for described to avoid mentioning twice words related to characteristics

·   Table 3, homologize the numbers’ punctuation to decimal points, there are commas and points

·       Line 261-262, in silico must be italic

·       Line 265, in dimethylthiazol-2-yl]-2,5-diphenyl tetrazolium the “l” is missed

·   Line 525, in 3-(4.5-dimethylthiazol-2-yl)-2.5-diphenyltetrazolium the points must be commas

Author Response

The authors thank all the reviewers for the modifications and suggestions, which contribute for the improvement of the original manuscript.

Reviewer 1:

Answers: The following suggestions/modifications were done in the manuscript.

  • Scheme 1,in a 3h must be 4h.
  • Line 134,where must be were
  • Table 2,check conventional heating values for 3d
  •   Line 155, I suggest changing characterized for described to avoid mentioning twice words related to characteristics
  • Table 3, homologize the numbers’ punctuation to decimal points, there are commas and points
  • Line 261-262, in silico must be italic
  • Line 265, in dimethylthiazol-2-yl]-2,5-diphenyl tetrazolium the “l” is missed
  •  Line 525, in 3-(4.5-dimethylthiazol-2-yl)-2.5-diphenyltetrazolium the points must be commas

Reviewer 2 Report

1. Page 1 line 20 and 21, page 10 line 302 inhibition50µM should be changed as IC50

2. The author should explain why the MA strategy were not applied to produce the compounds 4a-d and 4g. (The 4-substituted-1,2-NQ derivatives (4a-d and 92 4g) were obtained as previously reported by a biocatalytic strategy using CotA-laccase, 93 from Bacillus subtilis)

3. The author make a correction on the Scheme 1 since there is no compound 3h, it should be 4h.

4. The author should explain why the compound 3h was not synthesized. 

5. In silico studies section, the author should provide an extended explanation on the reason why the compounds are target for kinase enzymes

6. The author should make a logical explanation why the HepG2 cell lines were used for cytotoxic activity.

7. The compounds were only conducted for cytotoxic effect. The toxicity study must also be done against the healthy cells to be able evaluate as good anti-cancer agents. Toxicity study is crucial for anti-cancer studies.  

8. In the inhibition of acetylcholinesterase study, the chosen concentration 50uM is not meaningful compared with the standards. The IC50 concentrations for the potent compounds (30 and 16uM) are also not valuable for the AChE inhibitions. That’s why inhibition of acetylcholinesterase for these compounds are not meaningful.

9. The author should make a correlation between cytotoxic evaluation and acetylcholinesterase inhibition profiles of the compounds. 

Author Response

The authors thank all the reviewers for the modifications and suggestions, which contribute for the improvement of the original manuscript.

Reviewer 2:

  1. Page 1 line 20 and 21, page 10 line 302 inhibition50µMshould be changed as IC50

Answer:

The percentages of inhibition shown were obtained for a concentration of 50 mM of each compound under study and therefore do not correspond to the IC50 for the same compound, as referred in lines 554 to 557.

  1. The author should explain why the MA strategy were not applied to produce the compounds 4a-d and 4g. (The 4-substituted-1,2-NQ derivatives (4a-d and 92 4g) were obtained as previously reported by a biocatalytic strategy using CotA-laccase, 93 from Bacillus subtilis)

Answer:

The 4-substituted-1,2-NQ derivatives 4a-d and 4g had already been synthesized by our group within our strategy of using biocatalysts to produce naphthoquinone derivatives and published in the literature [reference 31 of the manuscript]. The MA strategy was applied only to compounds, which were not obtained by biocatalysis, and as a proof of concept that this approach can also be applied to the 1,2-NQ family of compounds.   

  1. The author make a correction on the Scheme 1 since there is no compound 3h, it should be 4h.

Answer: Corrected.

  1. The author should explain why the compound 3h was not synthesized. 

Answer:

Compound 4h was synthetised to evaluate the importance of the hydroxyl position in the aromatic ring of nucleophile when compared with 4b, one of the most potent compounds studied in this series. The results showed that only a slight improvement was achieved. Since the homologue in the 1,4-NQ family was not relevant as AChE inhibitor we did not consider its synthesis.

  1. In silico studies section, the author should provide an extended explanation on the reason why the compounds are target for kinase enzymes

Answer:

As requested by the reviewer it was introduced one sentence evidencing the binding interactions between 1,4-NQs and the kinases catalytic sites already reported in the literature. The following text was introduced:

“… (see Table S5). In fact, the 1,4-NQ skeleton and other quinone derived compounds had already been reported as active against kinases, by biochemical binding and molecular interaction with the kinases catalytic sites [39]. (lines 249 to 252).

  1. The author should make a logical explanation why the HepG2 cell lines were used for cytotoxic activity.

Answer:

HepG2 is a hepatic cell line that has been used in a wide range of studies, often to demonstrate the anti-cancer potential of several types of compounds, enabling the comparison of the relevance of compounds under study. Furthermore, this cell line is related with to liver cancer, which is the third leading cause of cancer death worldwide, with most cases of hepatocellular carcinoma, and HepG2 is a model cell line for this type of cancer. A sentence and a new reference [41] were introduced in the text according to the reviewer suggestion (lines 263-264).

  1. The compounds were only conducted for cytotoxic effect. The toxicity study must also be done against the healthy cells to be able evaluate as good anti-cancer agents. Toxicity study is crucial for anti-cancer studies.  

Answer:

We agree with the reviewer. However, this work presents a screening preliminary approach, aiming to evaluate the potential of these naphthoquinone derivatives as anticancer agents. Our results already showed which of the compounds are most promising for future evaluation. We would like to thank the reviewer suggestion and agree that this future evaluation should include toxicity studies on human primary cell lines from healthy tissues, as well as the effect against other type of cancers cell. 

  1. In the inhibition of acetylcholinesterase study, the chosen concentration 50uM is not meaningful compared with the standards. The IC50concentrations for the potent compounds (30 and 16uM) are also not valuable for the AChE inhibitions. That’s why inhibition of acetylcholinesterase for these compounds are not meaningful.

Answer:

We agree with the reviewer that our compounds are only moderate AChE inhibitors and we changed the manuscript accordingly. The new sentences introduced are:

“…These values, higher than the AChEIC50 values of the reference AChE inhibitors rivastigmine (1.03 μM) or galantamine (1.99 μM)[46], show that these compounds are only moderate AChE inhibitors and further structural changes must be considered for these 1,2-NQ and 1,4-NQ families can be foreseen as alternative drug scaffolds.” (lines 322 to 326).

  1. The author should make a correlation between cytotoxic evaluation and acetylcholinesterase inhibition profiles of the compounds. 

Answer:

We thank the reviewer suggestion.The aim of this work was to study the potential of these families of compounds as biologically active compounds. The targets and mechanisms associated with each of assayed biological activities are complex and may have completely different origins, therefore a more deepen study should be performed as a future perspective.

However, in the conclusion section the text was modified as:

“…The potential biological activity for the compounds was assessed for their cytotoxicity against HepG2 cell lines and as AChE inhibitors, showing that 1,2-NQ series exhibit a dual action, both as promising cytotoxic agents and moderate AChE inhibitors. The results showed a dependence on chemical modifications in the NQ moiety with an increase on cytotoxic and the AChE inhibitory effects, in particular for electron-donating substituted aromatic amines...”  (lines 576 to 581)

Reviewer 3 Report

The manuscript entitled “Sustainable synthesis, antiproliferative and acetylcholinesterase inhibition of 1,4- and 1,2-naphthoquinone derivatives” aims to describe the design, sustainable synthesis, evaluation of electrochemical and biologic properties against HepG2 cell lines and AChE enzyme of different substituted derivatives of 1,4- and 1,2-naphthoquinones. A microwave-assisted protocol was successfully optimized for synthesizing the 2-substituted-1,4-NQ series and extended to the 4-substituted-1,2-NQ family. This topic is significant. The manuscript is very well prepared. The content is suitable for the Journal. I have some minor comments, but it is essential to address them. The comments are given below.

Lines 53 and 54: “The drugs currently available in the AD therapy, such as the AChEIs donepezil, galantamine and rivastigmine have shown modest benefits [23]” – the authors should be careful with the statements like this. Inhibitors of AChE are not problematic because of their modest effect. They treat the symptoms quite well but do not cure the disease. Also, their side effects are severe.  

Lines 71 – 74: “Microwave assisted (MA) organic synthesis is considered an important green approach and, nowadays, is widely used in the pharmaceutical industry for the oriented synthesis of target compounds [29, 30], allowing for cleaner reactions, in shorter times and with higher yields.” It is not clear why MA synthesis is considered green and sustainable. Shorter time and higher yields are hardly the reasons. The authors must elaborate on this better. 

Author Response

The authors thank all the reviewers for the modifications and suggestions, which contribute for the improvement of the original manuscript.

Reviewer 3:

Lines 53 and 54: “The drugs currently available in the AD therapy, such as the AChEIs donepezil, galantamine and rivastigmine have shown modest benefits [23]” – the authors should be careful with the statements like this. Inhibitors of AChE are not problematic because of their modest effect. They treat the symptoms quite well but do not cure the disease. Also, their side effects are severe.  

 Answer:

We agree with the reviewer, and we changed the manuscript accordingly. The new sentences introduced are: “The drugs currently available, such as the AChEIs donepezil, galantamine and rivastigmine, used in the symptomatic treatment of AD, show non-negligible adverse side effects …” (lines 53 to 55).

and

“… The adverse effects shown by AChEIs in use in the treatment of Alzheimer and dementia symptons [23],…” (lines 298 to 299)

Lines 71 – 74: “Microwave assisted (MA) organic synthesis is considered an important green approach and, nowadays, is widely used in the pharmaceutical industry for the oriented synthesis of target compounds [29, 30], allowing for cleaner reactions, in shorter times and with higher yields.” It is not clear why MA synthesis is considered green and sustainable. Shorter time and higher yields are hardly the reasons. The authors must elaborate on this better. 

Answer: Following the Reviewer’s suggestion we changed the manuscript accordingly. The new text introduced is: “… allowing for more energy efficient and cleaner reactions, the choice of green solvents, time-efficient routes and higher yields.” (Lines 73-74).

Reviewer 4 Report

The present manuscript of Robalo et al. reports the synthesis, NMR- and mass-spectral characterization of a series of 2-substituted 1,4-naphthoquinones and 4-substituted 1,2-naphthoquinones, the investigation of their electrochemical properties (by means of cyclic voltammetry) and evaluation of some biochemical properties of these quinones. In general, I have found this manuscript to be interesting and well-written. The experimental part was performed in appropriate level and causes no doubts. The optimal conditions for a microwave-assisted synthesis were found and applied for the preparation of a number of substituted 1,4- and 1,2-naphthoquinones. The authors have proven the structures of a series of naphthoquinones using 1H and 13C NMR spectroscopy, as well as by ESI-MS and  ESI-HRMS. 

In addition to the "in vitro" cytotoxicity against human hepatocarcinoma cell line HepG2 and inhibition of acetylcholinesterase activity, the molecular docking studies were also provided.

The conclusions are completely supported by the experimental results. I believe this manuscript will make a good impact to the area of biochemical investigations of quinone type compounds. 

Before the acceptance of the manuscript, I would like to pay the attention of the authors to the following minor moments:

(1) Part 2.2. Electrochemistry Studies.

The Fig. 3 shows the CV of 1,4-naphthoquinone, 1.2-naphthoquinone and only one naphthoquinone 3a from the series described in the manuscript. I would prefer to see the figures with CV of studied substituted naphthoquinones than of the well-known parent 1,4- and 1.2-naphthoquinones. Also, it would be very helpful for readers if authors give CVs for all studied quinones in ESI. 

(2) Part 4 (Experimental details).

Line 428: 100 m.Vs-1  should be 100 mVs-1

Lines 444, 453, 461, 483, 490, 498, and 505: for 13C NMR the frequency is 100 MHz but not 400 MHz. Please check.

Lines 466-470. The 13C NMR spectral data for 3d are absent. Please check.

(3) Supplementary Materials. 

Line 597. "...Figures S4 to S37..." should be "...Figures S4 to S36..."; "...Figures S38 to 45..." should be "...Figures S38 to S45..."

(4) File ESI

1. The titles of figures should be on the same page as the figures. Please pay attention to Figures S2 ans S3.

2. The reference electrode SCE should be mentioned in titles to Tables S1 and S2, e.g. "Table S1. Electrochemical parameters for 1,4-NQ and derivatives in acetonitrile (potentials are given vs SCE; scan rate 0.1V.s-1)."

Good luck!

Author Response

The authors thank all the reviewers for the modifications and suggestions, which contribute for the improvement of the original manuscript.

Reviewer 4:

(1) Part 2.2. Electrochemistry Studies.

The Fig. 3 shows the CV of 1,4-naphthoquinone, 1.2-naphthoquinone and only one naphthoquinone 3a from the series described in the manuscript. I would prefer to see the figures with CV of studied substituted naphthoquinones than of the well-known parent 1,4- and 1.2-naphthoquinones. Also, it would be very helpful for readers if authors give CVs for all studied quinones in ESI. 

Answer:

CVs of the 1,4- and 1,2-naphthoquinone families have been introduced in the supplementary materials file (Figures S2 to S14) 

Lines 466-470. The 13C NMR spectral data for 3d are absent. Please check.

Answer:

The 13C data for compound 3d was not presented because despite all attempts to accumulate the 13C spectrum, it was not possible to obtain a spectrum with sufficient quality to extract all 13C data. However, the structure of the compound was confirmed through the exact mass.

(2) Part 4 (Experimental details).

Answer:

All the following modifications suggested by the reviewer were included in the revised manuscript.

Line 428: 100 m.Vs-1  should be 100 mVs-1

Lines 444, 453, 461, 483, 490, 498, and 505: for 13C NMR the frequency is 100 MHz but not 400 MHz. Please check.

(3) Supplementary Materials. 

Line 597. "...Figures S4 to S37..." should be "...Figures S4 to S36..."; "...Figures S38 to 45..." should be "...Figures S38 to S45..."

(4) File ESI

  1. The titles of figures should be on the same page as the figures. Please pay attention to Figures S2 ans S3.
  2. The reference electrode SCE should be mentioned in titles to Tables S1 and S2, e.g. "Table S1.Electrochemical parameters for 1,4-NQ and derivatives in acetonitrile (potentials are given vsSCE; scan rate 0.1V.s-1)."

Round 2

Reviewer 2 Report

Accepted